# Patient Judgement of Change with Elective Surgery Correlates with Patient Reported Outcomes and Quality of Life

**DOI:** 10.3390/healthcare10060999

**Published:** 2022-05-27

**Authors:** Meg E. Morris, Victoria Atkinson, Jeffrey Woods, Paul S. Myles, Anita Hodge, Cathy H. Jones, Damien Lloyd, Vincent Rovtar, Amanda M. Clifford, Natasha K. Brusco

**Affiliations:** 1La Trobe University Academic and Research Collaborative in Health (ARCH), Bundoora 3083, Australia; victoria.atkinson@healthscope.com.au (V.A.); jeffrey.woods@healthscope.com.au (J.W.); anita.hodge@healthscope.com.au (A.H.); cathyjoneshealth@gmail.com (C.H.J.); damien.lloyd@healthscope.com.au (D.L.); natasha.brusco@monash.edu (N.K.B.); 2Victorian Rehabilitation Centre, Healthscope Limited, Glen Waverley, Bundoora 3083, Australia; 3Healthscope Limited, Melbourne, Bundoora 3083, Australia; vincent.rovtar@healthscope.com.au; 4Anaesthesiology and Perioperative Medicine, Alfred Hospital and Monash University Central Clinical School, Melbourne, Bundoora 3083, Australia; p.myles@alfred.org.au; 5School of Allied Health, Ageing Research Centre, Health Research Institute, University of Limerick, V94 Limerick, Ireland; amanda.clifford@ul.ie; 6Alpha Crucis Group, Bundoora 3083, Australia; 7Rehabilitation, Ageing and Independent Living (RAIL) Research Centre, Monash University Melbourne, Bundoora 3083, Australia

**Keywords:** patient reported outcome measure (PROM), quality of life (QOL), quality of recovery (QOR), hospital, implementation science, hospital, surgery

## Abstract

Obtaining pre-surgery PROM measures is not always feasible. The aim of this study was to examine if self-reports of change following elective surgery correlate with change scores from a validated PROM (15-item Quality of Recovery (QoR-15)). This cross-sectional study across 29 hospitals enrolled elective surgery patients. PROMs were collected one-week pre-surgery, as well as one- and four-weeks post-surgery via an electronic survey. We examined associations between patient “judgement of change” at one and four-weeks after surgery and the actual pre-to post-surgery PROM change scores. A total of 4177 surveys were received. The correlation between patient judgement of change, and the actual change score was moderately strong at one-week (n = 247, rs = 0.512, *p* < 0.001), yet low at four-weeks (n = 241, rs = 0.340, *p* < 0.001). Patient judgement was aligned to the direction of the PROM change score from pre- to post-surgery. We also examined the correlation between the QoR-15 (quality of recovery) and the EQ-5D-5L (QOL). There was a moderately strong positive correlation between the two PROMs (n = 356, rs = 0.666, *p* < 0.001), indicating that change in quality of recovery was related to change in QOL. These findings support the use of a single “judgement of change” recall question post-surgery.

## 1. Introduction

The patient voice is essential for identifying patient goals, for ensuring meaningful patient-clinician communication and patient centred car, and a critical step to hearing the patient voice in healthcare is the integration of patient reported outcome measures (PROMs) into routine care [1,2]. Despite the benefits of routine PROM use in clinical care, barriers still exist and PROM utilisation is not consistent [3,4]. Specific to elective surgery, the inclusion of PROMs, alongside other key outcomes such as hospital length of stay, re-admissions, complications and mortality, is important in measuring surgical risks and outcome [5]. However, much is unknown about how to streamline the PROM implementation process across complex health networks, to minimise the burden on the patient and the hospital [6], such as achieving a pre-surgery PROM score when it is not feasible to obtain this prior to surgery [7].

PROMs are usually completed by patients, or their family members to gain information on outcomes in real time [8]. PROMs can also be collected over multiple time points to understand change over time [9]. Some emergency surgical patients are not able to complete a PROM prior to surgery due to the nature of an emergency presentation. Therefore, it is common to ask the patient to recall the PROM results based on a previous point in time. The validity of asking patients to recall their PROMs is uncertain. Caution needs to be exercised with this approach as prior studies have failed to reach consensus on the accuracy of recalled PROM scores compared to PROM scores reported in real time, as several have reported lower recalled scores [10,11,12].

Both the 15-item Quality of Recovery 15 (QoR-15) and EuroQOL 5-Dimension 5-Level quality of life (EQ-5D) [13,14] tools are examples of PROMs that are frequently used with surgical patient cohorts in the short and longer-term [6,15,16,17,18,19]. Several questions relating to these two tools remain unanswered. In particular, it is not known for elective surgery patients whether a 5-point better-worse scale on a single post-surgery recall question correlates with the QoR-15 pre- to post-surgery change score. It is also unclear whether change in quality of recovery is related to change in quality of life following elective surgery, using the QoR-15 and EQ-5D.

This nationwide PROM implementation study examined self-reported PROM data from patients undergoing elective surgery at 29 Australian hospitals. Patients completed PROM surveys before and after surgery, via an electronic survey. After surgery they also answered a single recall question about their “judgement of change”, specifically whether they were better or worse than pre-surgery (5-point scale of better or worse than prior to surgery). The primary aim was to determine the correlation and direction between patient judgements of change at one and four-weeks after surgery and the actual pre-to post-surgery PROM change scores. The secondary aim was to examine the correlation between the QoR-15 quality of recovery PROM and the EQ-5D-5L quality of life PROM, to report if changes in quality of recovery were related to changes in quality of life.

## 2. Materials and Methods

The study has been reported in accordance with the COSMIN reporting guideline for investigations on measurement properties of PROMs [20]. The COSMIN guidelines aim to improve the reporting transparency of studies investigating measurement properties of existing PROMs [20]. There are 71 items and this includes 35 common recommendations for all studies as well as an additional 36 recommendations for specific study types [20]. This study is part of a larger study to develop Australian ePROM implementation recommendations, called ‘AusPROM’ [6].

### 2.1. Participants

Across 29 hospitals patients who had elective surgery were eligible to participate in this study. Participants were categorized as an overnight patient, or as a day surgery patient. The detailed inclusion and exclusion criteria and full methods have been published previously [6]. In summary, participants were aged 18+ and were scheduled to have, or had, elective surgery at one of the 29 included hospitals. Participants were excluded if the surgery was diagnostic or obstetric.

### 2.2. Data Collection and Sample Size

Electronic PROMs were collected over a three months period, commencing September 2021. Data collection included three surveys which were distributed the week prior to elective surgery (Survey I), as well as one- and four-weeks following surgery (Survey II and III, respectively). Surgical lists informed the survey distribution list, with potential participants with excluded surgical types removed from the list prior to survey distribution.

The sample size was based on a 30% response rate and on achieving national hospital representation, across the 29 hospitals, with an estimated 15,000 surveys returned over three months (combining Surveys I, II, and III). For the correlation analysis, based on a two-tailed alpha of 0.05 and a correlation coefficient of 0.20, the total sample size required was 194 [21], which was far exceeded, achieving national hospital representation.

All hospitals completed the QoR-15, and three hospitals also completed the EQ-5D to address the secondary aim. As the individual hospitals could choose survey distribution method, via email or via short message service (SMS) on a smart phone, this data was collected. The response rate for each distribution method was reported, in addition to an overall response rate.

### 2.3. Outcome Measures

The QoR-15 tool is a generic PROM designed to report on quality of surgical recovery and has 15 items scored on a 0–10 scale with a maximum score of 150 and a minimal clinical important difference of 8 [18,19]. The EQ-5D is a generic PROM designed to report on quality of life and has five domains scored across five levels; the domains include mobility personal care, usual activities, pain/discomfort and anxiety/depression. Raw scores are converted into a utility index between 0 (a state equal to death) and 1 (perfect health) and the minimal clinical important difference is between 0.03 and 0.06 [13,14]. Both of these PROMs have been validated across multiple patient cohorts and in multiple languages, and take less than 3 min to complete [18,19,22,23,24].

Sample size for the analysis was based on a sample of convenience, which had been reported for the broader study [6]. Patient participants provided written consent via a tick box, prior to commencing each of the PROM surveys.

### 2.4. Statistical Analysis

Scores for the QoR-15 and the EQ-5D utility index have been presented as pooled means and standard deviations. The mean difference scores and associated 95% CIs were calculated between the Survey I, II and III results, and this analysis included all participants. In the two post-surgery surveys, patients were asked to recall their judgement of change on a five-point Likert scale (1 = Worse; 2 = A little worse; 3 = Much the same; 4 = A little better; 5 = Better). Correlation between the Likert scale and the pre- to post-surgery QoR-15 change scores were analysed in a paired fashion using Spearman’s rho coefficients [25], and this analysis only included participants who completed multiple surveys, i.e., Survey I in addition to Survey II and/or III. The Likert scale and the pre- to post-surgery QoR-15 change scores were also presented graphically to display the direction of QoR-15 change scores for each of the five-points on the Likert scale.

The correlations between the QoR-15 and the EQ-5D scores were also examined using a Spearman’s rho analysis for participants who had completed both the QoR-15 and the EQ-5D. As both the QoR-15 and the EQ-5D utility index were based on combining a number of individual domains, we also examined the correlation between similar domains when they were represented in both the QoR-15 and the EQ-5D, for example both measures have a domain on usual activities. A correlation of 0.0 indicated no correlation, 0.1 to 0.3 is a weak correlation, 0.4 to 0.6 a moderate correlation, 0.7 to 0.9 a strong correlation and 1.0 a perfect correlation [25,26]. Missing data were excluded case wise and there were no post hoc analyses. Statistical significance was determined at *p* < 0.05. Analyses were completed using SPSS Version 27 [27].

## 3. Results

Across all hospitals, a total of 4177 surveys were received (overall response rate 15.4%, (n = 4177/27,084), with 1135 from the week prior to surgery (Survey I), 1698 from the week following surgery (Survey II), and 1344 from four weeks following surgery (Survey III). The number of participants was less then expected due to the low response rate and due to the impact of the COVID-19 pandemic, e.g., cancelled elective surgery. Patient characteristics and response rates are detailed in Table 1. The 4177 survey responses represented 692 different surgical Medicare codes with the most frequent being Code 49580 Partial meniscectomy of the knee (n = 125; 3.0%), Code 46363 Tendon sheath of hand/wrist (n = 86; 2.1%) and 42702 Lens extraction and insertion of intraocular lens. Overnight and day surgery survey responses had 378 and 424 different surgical Medicare codes, respectively.

There were 2913 individual patients who completed the surveys and of these 1936 completed 1 survey, 703 completed 2 surveys, and 274 patients completed three or more surveys (more than three surveys indicate two or more separate surgeries for the one patient).

Including patients with complete PROM data for Survey I, II and/or III, QoR-15 scores significantly reduced from the week prior to surgery to the week following surgery (MD −8.31; 95% CI −10.28 to −6.35; *p* < 0.001), then significantly increased from the week following surgery to four weeks following surgery (MD 10.49; 95% CI 8.64 to 12.33; *p* < 0.001) (Table 2). Similarly, the EQ-5D scores significantly reduced from the week prior to surgery to the week following surgery (MD −0.068; 95% CI −0.124 to −0.011; *p* = 0.019), then significantly increased from the week following surgery to four weeks following surgery (MD 0.083; 95% CI 0.036 to 0.130; *p* = 0.001)) (Table 2). It was also observed that the QoR-15 and EQ-5D scores differed between the overnight stay admissions and the day admission, with the day admission scores higher on all three survey occasions (Table 2).

Including patients with complete PROM and judgement of change data, the correlation between the patient judgement of change response, and the actual change score between the pre-surgery score and post-surgery PROM score was moderately strong at one-week (n = 247, rs = 0.512, *p* < 0.001), but was low at four-weeks (n = 241, rs = 0.340, *p* < 0.001), indicating patient judgement of change was more aligned to the PROM change score at one-week post-surgery compared to four weeks post-surgery.

The direction of change between the patient judgement of change response, and the actual change score between the pre-surgery score and post-surgery QOR-15 PROM score is presented in Figure 1 and Figure 2 and Table 3.

There was a moderately strong positive correlation between the QoR-15 and the EQ-5D (n = 356, rs = 0.666, *p* < 0.001) across all participants (Figure 3). There was a strong positive correlation for Survey I and Survey III results (n = 76, rs = 0.733, *p* < 0.001; n = 120, rs = 0.702, *p* < 0.001; respectively), and a moderate positive correlation for Survey II results (n = 160, rs = 0.548, *p* < 0.001). Individual domains present in both the QoR-15 and the EQ-5D were all significantly correlated and displayed correlation values ranging from −0.414 to −0.766 (Table 4).

## 4. Discussion

The results of this investigation highlight the value of engaging consumers in the co-design of health service interventions. It particularly showed that elective surgery patients are capable of reliably reporting their health status before surgery and in the week after surgery, as well as the amount of change in health-related quality of life associated with surgical interventions, using electronic surveys. The results support the findings of global studies showing that it is practical and helpful to collect data on patient views on the value of intervention, alongside post-surgery morbidity and mortality [28,29,30]. Our results are also in agreement with Stirling Bryan et al. [30] who noted the need to balance collecting regular patient PROM feedback using procedures that are not too burdensome [29].

Using patient judgements of health status and quality of life before surgery has the potential to save healthcare organizations considerable time and resources in PROM administration. There is evidence to support the recall of pre-surgery health status by patients after they have received surgery or another therapy intervention [7,31,32,33]. For example, Kwong and Black (2017) conducted an evaluation of retrospectively reported PROMS in emergency department patients [31]. There were strong associations between retrospective and real-time PROM scores in 21 of 30 comparisons. The overall finding was that PROM recall is good in patients who are cognitively intact and medically stable. While retrospective PROM data collection offers one method to measure consumer views on changes in their outcomes over time, the current study showed the value of using a single recall question to report the patient judgement of change as a measure of change from pre- to post-surgery.

The main strength of this investigation was the national approach with 29 Australian hospitals contributing to the data collection. The main limitation was that the survey response rate was lower than expected due to the COVID-19 pandemic and associated lock-down and other public health measures. Specifically, Australian hospitals located in Victoria and New South Wales, which represented 19 of the 29 sites, were impacted by a wave of COVID-19, which resulted in reduced elective surgery as well as a “code red” classification by the project team The “code red” meant that Survey I was stopped for the majority of the data collection period in the 19 affected hospitals, due to the human resource demands associated with the partly automated/partly manual process; however, there was no change to Surveys II and III as these were fully automated. In addition, we were only able to administer surveys to people who could respond in English and who had access to computer, tablets or telephone platforms. Generalisability of this study is limited to the Australian context and to elective surgery patients. It is also noted that age is an independent variable in determining quality of life [34]. In the current study there was a wide age representation from 18 to 85+. While this age variation may limit the internal validity of the study, the age bands were evenly represented across Surveys I, II and III, minimizing the impact of age on the findings.

As with the ISOQOL recommendation for minimum standards for PROMs, it is acknowledged that this report only presents an early analysis of the judgement of change recall question and that future research is needed to investigate additional psychometric properties of the judgement of change recall question, such as validity, reliability and sensitivity to change over time [35]. In addition, future research is needed to determine whether these findings apply to patients with a diverse range of cultural backgrounds, ages and diagnoses. There is also a need to better understand how digital literacy contributes to the success of PROM data collection in elective surgery patients.

### Implications for Practice

When collecting PROMs with surgical patients, it is acceptable to use a single “judgement of change” recall question completed by patients one week after the surgery, to measure their views on the amount of change experiencedBefore and after elective surgery, it may not be essential to administer both a quality of recovery PROM and a quality of life PROM, as they are closely correlated

## 5. Conclusions

In the week following surgery, patients were able to reliably judge changes in their surgery outcomes. In the month following surgery, change in quality of recovery after surgery was related to change in quality of life. These findings support the use of a single “judgement of change” recall question post-surgery, to indicate change in health status over time.

## Figures and Tables

**Figure 1 healthcare-10-00999-f001:**
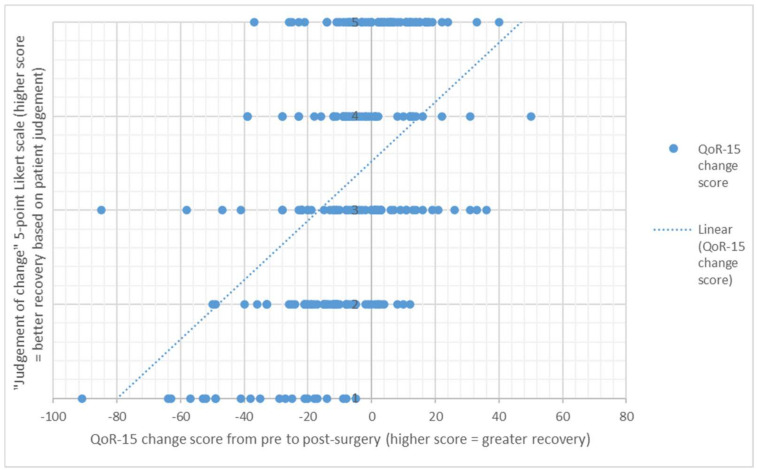
From pre to one week post-surgery, scatter plot of the QoR-15 change score and the “judgement of change” score on the 5-point Likert scale, n = 247 (1 = Worse; 2 = A little worse; 3 = Much the same; 4 = A little better; 5 = Better).

**Figure 2 healthcare-10-00999-f002:**
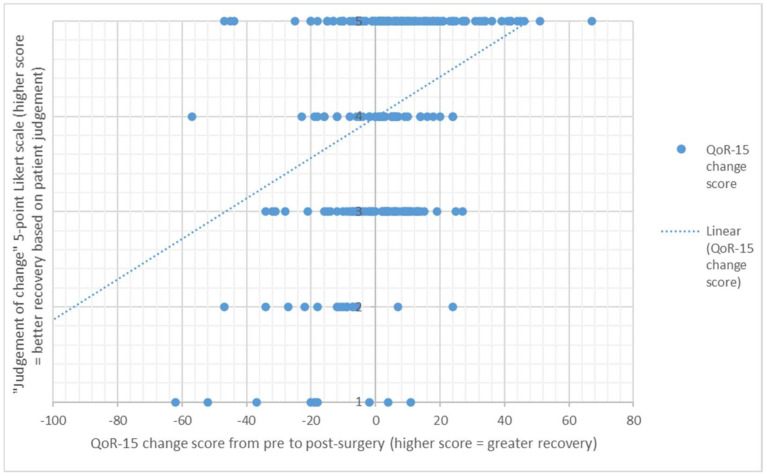
From pre to four weeks post-surgery, scatter plot of the QoR-15 change score and the “judgement of change” score on the 5-point Likert scale, n = 241 (1 = Worse; 2 = A little worse; 3 = Much the same; 4 = A little better; 5 = Better).

**Figure 3 healthcare-10-00999-f003:**
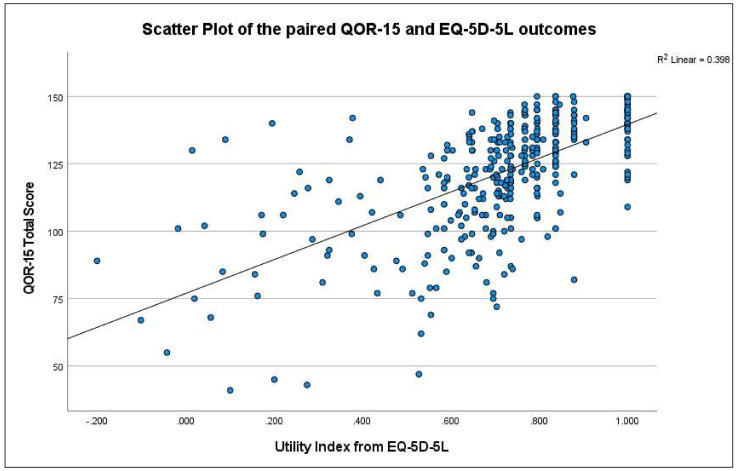
Scatter plot of the paired QOR-15 and EQ-5D outcomes.

**Table 1 healthcare-10-00999-t001:** Response rate and patient characteristics for surveys I, II and III.

		Survey I n = 1135	Survey II n = 1698	Survey III n = 1344
Email response rate	Sent	6263	9615	9612
	Responses	1063	1449	1197
	Response Rate	17.0%	15.1%	12.5%
Short Message Service (SMS) response rate	Sent	114	740	740
	Responses	72	249	147
	Response Rate	63.2%	33.6%	19.9%
Overall responses rate	Sent	6377	10,355	10,352
	Responses	1135	1698	1344
	Response Rate	17.8%	16.4%	13.0%
Age range		18–40, 244 (21.5%) 41–64, 490 (43.2%) 65–74, 274 (24.2%) 75–84, 119 (10.5%) 85+, 6 (0.5%)	18–40, 307 (18.1%) 41–64, 785 (46.4%) 65–74, 417 (24.6%) 75–84, 177 (10.5%) 85+, 7 (0.4%)	18–40, 212 (15.8%) 41–64, 613 (45.6%) 65–74, 357 (26.6%) 75–84, 154 (11.5%) 85+, 7 (0.5%)
Gender, female		673 (59.4%)	945 (55.8%)	733 (54.6%)
Day surgery		253 (47.3%)	938 (55.5%)	732 (54.6%)

**Table 2 healthcare-10-00999-t002:** QoR-15 and EQ-5D scores.

	Survey I	Survey II	Survey III	Mean Difference (95% CI, *p* Value)		
				Survey II Minus Survey I	Survey III Minus Survey II	Survey III Minus Survey I
QoR-15 *						
Overnight stay admissions	122.93 (SD 19.54; n = 188)	110.94 (SD 23.05; n = 540)	124.52 (SD 20.59; n = 473)	**−12.00 (−15.69 to −8.31; *p* < 0.001)**	**13.59 (10.88 to 16.30; *p* < 0.001)**	1.59 (−1.84 to 5.03; *p* = 0.363)
Day admissions	126.41 (SD 16.82; n = 165)	120.37 (SD 21.52; n = 633)	128.63 (SD 20.02; n = 497)	**−6.04 (−9.60 to −2.50; *p* = 0.001)**	**8.26 (5.81 to 10.72; *p* < 0.001)**	2.23 (−1.18 to 5.63; *p* = 0.199)
Combined overnight and day admissions	124.40 (SD 19.02; n = 738)	116.08 (SD 22.70; n = 1,178)	126.57 (SD 20.43; n = 973)	**−8.31 (−10.28 to −6.35; *p* < 0.001)**	**10.49 (8.64 to 12.33; *p* < 0.001)**	**2.18 (0.28 to 4.07; *p* = 0.025)**
EQ-5D ^#^						
Overnight stay admissions	0.634 (SD 0.295; n = 9)	0.616 (SD 0.260; n = 55)	0.735 (SD 0.205; n = 60)	−0.018 (−0.208 to 0.172; *p* = 0.851)	**0.118 (0.032 to 0.204; *p* = 0.007)**	0.078 (−0.055 to 0.256; *p* = 0.201)
Day admissions	0.823 (SD 0.145; n = 12)	0.694 (SD 0.177; n = 123)	0.767 (SD 0.218; n = 79)	**−0.129 (−0.234 to −0.024; *p* = 0.016)**	**0.073 (0.018 to 0.128; *p* = 0.010)**	−0.056 (−0.185 to 0.074; *p* = 0.394)
Combined overnight and day admissions	0.738 (SD 0.233; n = 84)	0.670 (SD 0.209; n = 178)	0.753 (SD 0.211; n = 140)	**−0.068 (−0.124 to −0.011; *p* = 0.019)**	**0.083 (0.036 to 0.130; *p* = 0.001)**	0.015 (−0.044 to 0.075; *p* = 0.615)

* The number of QOR-15 scores may be less than the number of surveys returned, due to missing data associated with participants not completing the QoR-15 in full, and therefore being excluded from the analysis. ^#^ The number of Utility Index scores may be less than the number of surveys returned, due to missing data due to participants not completing the Utility Index questions in full, and therefore being excluded from the analysis. In addition, Utility Index numbers were low compared to QoR-15 numbers as only three sites collected the Utility Index data compared to 29 sites collecting QoR-15 data.

**Table 3 healthcare-10-00999-t003:** The direction of change between the patient “judgement of change” response, and the actual change score between the pre-surgery score and post-surgery QOR-15 PROM score.

Likert Scale	From Pre to One Week Post-Surgery		From Pre to Four Weeks Post-Surgery	
	Number	Mean (SD)	Number	Mean (SD)
1 = Worse	23	−38.7 (25.9)	9	−21.7 (24.8)
2 = A little worse	59	−13.6 (12.9)	15	−12.5 (16.5)
3 = Much the same	57	−5.7 (21.2)	57	−2.1 (21.5)
4 = A little better	44	−2.8 (16.9)	38	0.1 (15.1)
5 = Better	64	2.9 (13.9)	122	8.2 (18.2)

**Table 4 healthcare-10-00999-t004:** Correlation values for individual domains present in both the QoR-15 and the EQ-5D.

	Number	Correlation Coefficient ^#^	*p* Value
EQ-5D: Personal Care, and QoR-15: Personal toilet and hygiene	412	−0.414	*p* < 0.001
EQ-5D: Usual Activities, and QoR-15: Usual Activities	401	−0.717	*p* < 0.001
EQ-5D: Pain, and QoR-15: Moderate Pain	404	−0.677	*p* < 0.001
EQ-5D: Pain, and QoR-15: Severe Pain	400	−0.660	*p* < 0.001
EQ-5D: Anxiety/Depression, and QoR-15: Worried or Anxious	398	−0.718	*p* < 0.001
EQ-5D: Anxiety/Depression, and QoR-15: Sad or Depressed	398	−0.766	*p* < 0.001

^#^ Negative correlation coefficients as these EQ-5D questions have a 1–5 scale with lower score = better state, and these QoR-15 questions have a 10–0 scale with lower score = poorer state.

## Data Availability

The named authors on this study will have access to the final trial dataset. Individual patient level data will not be available for sharing at the conclusion of this study.

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
