# Peer review of "Patient Judgement of Change with Elective Surgery Correlates with Patient Reported Outcomes and Quality of Life"

_healthcare, 2022, doi:10.3390/healthcare10060999_

Round 1

Reviewer 1 Report

Thank you for an interesting article.

I have the following comments:-

  • Elective surgery is a wide concept, and it may include a wide variety of surgeries such as simple procedures and cosmetic surgeries and it also may include more severe conditions such as knee or hip replacement. This variety of procedures may contribute to the quality of life changes. Therefore, I recommend that the authors need to have a conceptual definition of “elective surgery” related to their study. It would be interesting to widely categorize the participants according to the type of elective surgeries.
  • The age range of the participants is wide 18-85+ years, some confounding variables may contribute to the quality of life in addition to elective surgeries in different age groups, such as co-morbidities with advanced age for example. The authors may need to debate more on this issue in the results and discussion section to verify the internal validity of this study.
  • It is not clear, what the implications of this study are? The authors need to clarify the implications of the study in the discussion section or in the conclusion.

Author Response

Reviewer: Thank you for an interesting article. I have the following comments:-

  • Elective surgery is a wide concept, and it may include a wide variety of surgeries such as simple procedures and cosmetic surgeries and it also may include more severe conditions such as knee or hip replacement. This variety of procedures may contribute to the quality of life changes. Therefore, I recommend that the authors need to have a conceptual definition of “elective surgery” related to their study. It would be interesting to widely categorize the participants according to the type of elective surgeries.

RESPONSE: This is a good point raised by the reviewer. To describe the diversity of surgery types included, the following has been added to the results section. “The 4,177 survey responses represented 692 different surgical Medicare codes with the most frequent being Code 49580 Partial meniscectomy of the knee (n=125; 3.0%), Code 46363 Tendon sheath of hand / wrist (n=86; 2.1%) and 42702 Lens extraction and insertion of intraocular lens. Overnight and day surgery survey responses had 378 and 424 different surgical Medicare codes, respectively.”

  • The age range of the participants is wide 18-85+ years, some confounding variables may contribute to the quality of life in addition to elective surgeries in different age groups, such as co-morbidities with advanced age for example. The authors may need to debate more on this issue in the results and discussion section to verify the internal validity of this study.

RESPONSE: This is a good point and does represent a potential limitation. The discussion has been expanded to state the following: “It is also noted that age is an independent variable in determining quality of life [30]. In the current study there was a wide age representation from 18 to 85+. While this age variation may limit the internal validity of the study, the age bands were evenly represented across Surveys I, II and III, minimizing the impact of age on the findings.” 

  • It is not clear, what the implications of this study are? The authors need to clarify the implications of the study in the discussion section or in the conclusion.

RESPONSE: The conclusion has been updated to state: “In the week following surgery, patients were able to reliably judge changes in their pre- to post-surgery outcomes. In the month following surgery, change in quality of recovery after surgery was related to change in quality of life. To obtain PROMs pre and post-surgery, these findings support the use of a single “judgement of change” recall question post-surgery, to indicate how the patient was feeling prior to surgery, when asking this question prior to surgery is not feasible.”

Reviewer 2 Report

Dear authors

I would like to thank you for giving me the opportunity to review the manuscript entitled “Patient judgment of change with elective surgery correlate with patient reported outcomes and quality of life”. The aim of this study was to examine if self-reports of change following elective surgery correlate with change scores from a validated patient reported outcome measure. This work is provided valuable information that can improve evidence-based practice in elective surgery patients. However, the manuscript should be carefully revised. My comments are as follows:

Abstract

  1. Please add what was done previously and what is needed to do (justify your research)?
  2. The conclusions of the study should be added.

Introduction

  1. You need to develop the introduction section and justify your research, for instance why patient judgment with an elective surgery is essential.

Methods

  1. Please add some content about the COSMIN reporting guideline.
  2. Although the detailed inclusion and exclusion criteria and full methods of your study have been published previously in the protocol study, I think you should add these items at least in summarized form.
  3. Please add complete data collection section and sample size calculation.
  4. Domains of EQ-5D should be added.

Results

  1. I think paragraph 2 results is suitable for the limitation section.

Discussion

  1. The discussion section should be developed and presented more interpretively rather than repeat the results.
  2. The results of the study what implication have for practice? add the implications for the practice section.
  3. Again in the conclusion section, you should state your study what messages have for patients and healthcare providers in addition to summarizing of key findings.

Author Response

Reviewer: Dear authors

I would like to thank you for giving me the opportunity to review the manuscript entitled “Patient judgment of change with elective surgery correlate with patient reported outcomes and quality of life”. The aim of this study was to examine if self-reports of change following elective surgery correlate with change scores from a validated patient reported outcome measure. This work is provided valuable information that can improve evidence-based practice in elective surgery patients. However, the manuscript should be carefully revised. My comments are as follows:

Abstract

  1. Please add what was done previously and what is needed to do (justify your research)?

RESPONSE: This has been briefly added, noting that the abstract has a 200 word limit. Abstract: “Obtaining pre-surgery PROM measures is not always feasible. The aim of this study was to examine if self-reports of change following elective surgery correlate with change scores from a validated PROM (15-item Quality of Recovery (QoR-15)). This cross-sectional study across 29 hospitals enrolled elective surgery patients. PROMs were collected one-week pre-surgery, as well as one- and four-weeks post-surgery via an electronic survey. We examined associations between patient “judgement of change” at one and four-weeks after surgery and the actual pre-to post-surgery PROM change scores. A total of 4,177 surveys were received. The correlation between patient judgement of change, and the actual change score was moderately strong at one-week (n=247, rs=0.512, p<0.001), yet low at four-weeks (n=241, rs=0.340, p<0.001). Patient judgement was aligned to the direction of the PROM change score from pre- to post-surgery. We also examined the correlation between the QoR-15 (quality of recovery) and the EQ-5D-5L (QOL). There was a moderately strong positive correlation between the two PROMs (n=356, rs=0.666, p<0.001), indicating that change in quality of recovery were related to change in QOL. These findings support the use of a single “judgement of change” recall question post-surgery, to indicate how the patient was feeling pre-surgery, when asking this question pre-surgery is not feasible.”

  1. The conclusions of the study should be added.

RESPONSE: This has been briefly added, noting that the abstract has a 200 word limit. See point above for updated abstract.

Introduction

  1. You need to develop the introduction section and justify your research, for instance why patient judgment with an elective surgery is essential.

RESPONSE: The following paragraph has been added to the introduction: “The patient voice is essential for identifying patient goals, for ensuring meaningful patient-clinician communication and patient centred care, and a critical step to hearing the patient voice in healthcare is the integration of patient reported outcome measures (PROMs) into routine care [1, 2]. Despite the benefits of routine PROM use in clinical care, barriers still exist and PROM utilisation is not consistent [3, 4]. Specific to elective surgery, the inclusion of PROMs, alongside other key outcomes such as hospital length of stay, re-admissions, complications and mortality, is important in measuring surgical risks and outcome [5]. However, much is unknown about how to streamline the PROM implementation process across complex health networks, to minimise the burden on the patient and the hospital [6], such as achieving a pre-surgery PROM score when it is not feasible to obtain this prior to surgery [7].  

Methods

  1. Please add some content about the COSMIN reporting guideline.

RESPONSE: Appendix 1 contains a full copy of the COSMIN reporting guidelines. At the start of the Appendix, we have added the following statement to further add content about the guidelines. “The COSMIN guidelines aim to improve the reporting transparency of studies investigating measurement properties of existing PROMs[14]. There are 71 items and this includes 35 common recommendations for all studies as well as an additional 36 recommendations for specific study types[14].“

  1. Although the detailed inclusion and exclusion criteria and full methods of your study have been published previously in the protocol study, I think you should add these items at least in summarized form.

RESPONSE: The methods have been expanded to state the following: “The detailed inclusion and exclusion criteria and full methods have been published previously [11]. In summary, participants were aged 18+ and were scheduled to have, or had, elective surgery at one of the 29 included hospitals. Participants were excluded if the surgery was diagnostic or obstetric.”

  1. Please add complete data collection section and sample size calculation.

RESPONSE: The following has been added to the methods: “Data collection and sample size: Electronic PROMs were collected over a three months period, commencing September 2021). Data collection included three surveys which were distributed the week prior to elective surgery (Survey I), as well as one- and four-weeks following surgery (Survey II and III, respectively). Surgical lists informed the survey distribution list, with potential participants with excluded surgical types removed from the list prior to survey distribution.

The sample size was based on a 30% response rate and achieving national hospital representation, across the 29 hospitals, with an estimated 15,000 surveys returned over three months (combining Surveys I, II, and III). For the correlation analysis, based on a two-tailed alpha of 0.05 and a correlation coefficient of 0.20, the total sample size required was 194 [15], which was far exceeded via achieving national hospital representation.“

  1. Domains of EQ-5D should be added.

RESPONSE: This has been added to the methods section.

Results

  1. I think paragraph 2 results is suitable for the limitation section.

RESPONSE: This has been moved as suggested.

Discussion

  1. The discussion section should be developed and presented more interpretively rather than repeat the results.

RESPONSE:.The discussion has been re-drafted based on your advice. Due to the significant changes, please refer to the revised manuscript.

  1. The results of the study what implication have for practice? add the implications for the practice section.

RESPONSE:  The following has been added to the end of the discussion: “Implications for practice:

  • When collecting PROMs with surgical patients, it is acceptable to use a single “judgement of change” recall question completed by patients one week after the surgery, to measure their views on the amount of change experienced
  • Before and after elective surgery, it may not be essential to administer both a quality of recovery PROM and a quality of life PROM, as they are closely correlated

  1. Again in the conclusion section, you should state your study what messages have for patients and healthcare providers in addition to summarizing of key findings.

RESPONSE: The conclusion has been updated to state: “In the week following surgery, patients were able to reliably judge changes in their pre- to post-surgery outcomes. In the month following surgery, change in quality of recovery after surgery was related to change in quality of life. To obtain PROMs pre and post-surgery, these findings support the use of a single “judgement of change” recall question post-surgery, to indicate how the patient was feeling prior to surgery, when asking this question prior to surgery is not feasible.” 

Reviewer 3 Report

Peer Review: Patient judgement of change with elective surgery correlate with patient reported outcomes and quality of life

Summary:

  • The authors report a multicentre investigation of the correlation between PROM scores and a single answer “judgement of change” from pre-op to the current time (1 week and 4 weeks post-op)
  • The authors found a moderately strong correlation at 1 week post-op and a poor correlation at 4 weeks post-op.
  • The authors conclude that patients can reliably judge their pre-op to 1 week post-op change in outcome/quality of recovery.

Strengths:

  • Well conducted study
  • Multicentre
  • Reasonable response rate given the circumstances (i.e., COVID)
  • Conservative and justified conclusions

Major Concerns:

  • It seems that these data should be analyzed in a paired fashion (i.e., comparing the same patient’s scores for Survey I, to Survey II and Survey III); however, it is not clear based on reading the methods section if this was done, or if the scores were analyzed as pooled means. Can this be clarified?

Minor Concerns:

  • Typographical error page 2, line 4, “In particular, is isnoto…”
  • Authors should add citation for the description of rs values as “strong”, “moderate” or “weak” – e.g., Moore, D. S., Notz, W. I, & Flinger, M. A. (2013). The basic practice of statistics (6th ed.). New York, NY: W. H. Freeman and Company. Page (138).; or Zikmund, William G. (2000). Business research methods (6th ed). Fort Worth: Harcourt College Publishers. (Page 513).

Author Response

Reviewer: Patient judgement of change with elective surgery correlate with patient reported outcomes and quality of life

Summary:

  • The authors report a multicentre investigation of the correlation between PROM scores and a single answer “judgement of change” from pre-op to the current time (1 week and 4 weeks post-op)
  • The authors found a moderately strong correlation at 1 week post-op and a poor correlation at 4 weeks post-op.
  • The authors conclude that patients can reliably judge their pre-op to 1 week post-op change in outcome/quality of recovery.

 Strengths:

  • Well conducted study
  • Multicentre
  • Reasonable response rate given the circumstances (i.e., COVID)
  • Conservative and justified conclusions

Major Concerns:

  • It seems that these data should be analyzed in a paired fashion (i.e., comparing the same patient’s scores for Survey I, to Survey II and Survey III); however, it is not clear based on reading the methods section if this was done, or if the scores were analyzed as pooled means. Can this be clarified?

RESPONSE: Thankyou for noting this important need for clarifications. Both pooled means and a paired fashion were used, for the different analyses. The methods have been updated to state:Scores for the QoR-15 and the EQ-5D utility index have been presented as pooled means and standard deviations. The mean difference scores and associated 95% CIs were calculated between the Survey I, II and III results, and this analysis included all participants. In the two post-surgery surveys, patients were asked to recall their judgement of change on a five-point Likert scale (1=Worse; 2=A little worse; 3=Much the same; 4=A little better; 5=Better). Correlation between the Likert scale and the pre- to post-surgery QoR-15 change scores were analysed in a paired fashion using Spearman’s rho coefficients [18], and this analysis only included participants who completed multiple surveys, i.e., Survey I in addition to Survey II and / or III.”

Minor Concerns:

  • Typographical error page 2, line 4, “In particular, is isnoto…”

RESPONSE: Thankyou, this has been corrected.

  • Authors should add citation for the description of rs values as “strong”, “moderate” or “weak” – e.g., Moore, D. S., Notz, W. I, & Flinger, M. A. (2013). The basic practice of statistics (6th ed.). New York, NY: W. H. Freeman and Company. Page (138).; or Zikmund, William G. (2000). Business research methods (6th ed). Fort Worth: Harcourt College Publishers. (Page 513).

RESPONSE: Good reference suggestions – the Moore reference has been added.

Round 2

Reviewer 1 Report

Thank you for taking my comments into consideration. I think that the manuscript has been improved. 

Reviewer 2 Report

Dear authors 

Thank you for addressing my comments.